# Unidirectional spin density wave state in metallic $(Sr_{1-x}La_x)_2IrO_4$

Xiang Chen [1,2], Julian L. Schmehr[2], Zahirul Islam[3], Zach Porter[4], Eli Zoghlin[2], Kenneth Finkelstein[5], Jacob P.C. Ruff[5] & Stephen D. Wilson[2]

Materials that exhibit both strong spin–orbit coupling and electron correlation effects are predicted to host numerous new electronic states. One prominent example is the $J_{eff} = 1/2$ Mott state in $Sr_2IrO_4$, where introducing carriers is predicted to manifest high temperature superconductivity analogous to the $S = 1/2$ Mott state of $La_2CuO_4$. While bulk superconductivity currently remains elusive, anomalous quasiparticle behaviors paralleling those in the cuprates such as pseudogap formation and the formation of a $d$-wave gap are observed upon electron-doping $Sr_2IrO_4$. Here we establish a magnetic parallel between electron-doped $Sr_2IrO_4$ and hole-doped $La_2CuO_4$ by unveiling a spin density wave state in electron-doped $Sr_2IrO_4$. Our magnetic resonant X-ray scattering data reveal the presence of an incommensurate magnetic state reminiscent of the diagonal spin density wave state observed in the monolayer cuprate $(La_{1-x}Sr_x)_2CuO_4$. This link supports the conjecture that the quenched Mott phases in electron-doped $Sr_2IrO_4$ and hole-doped $La_2CuO_4$ support common competing electronic phases.

[1] Department of Physics, Boston College, Chestnut Hill, MA 02467, USA. [2] Materials Department, University of California, Santa Barbara, CA 93106, USA. [3] Advanced Photon Source, Argonne National Laboratory, Argonne, IL 60439, USA. [4] Department of Physics, University of California, Santa Barbara, CA 93106, USA. [5] Cornell High Energy Synchrotron Source, Cornell University, Ithaca, NY 14853, USA. Correspondence and requests for materials should be addressed to S.D.W. (email: stephendwilson@ucsb.edu)

The interplay between strong crystal field splitting, strong spin–orbit coupling, and on-site Coulomb interactions can lead to the formation of a new form of Mott state—one where the quenching of orbital degeneracy by spin–orbit coupling gives rise to a half-filled band with a correlation-driven charge gap[1–4]. The resulting spin–orbit assisted Mott state readily forms in a number of $5d$ transition metal oxides with a key example being compounds with pentavalent $Ir^{4+}$ cations sitting in a locally cubic crystal field that stabilizes the formation of a $J_{eff} = 1/2$ spin–orbit entangled wave function. The $J_{eff} = 1/2$ Mott state of $Sr_2IrO_4$ in particular has drawn considerable interest due to electronic and structural parallels drawn between it and the structurally related $S = 1/2$ Mott state of $La_2CuO_4$[5–9]. Theoretical work mapping an effective single band Hubbard model for the $S = 1/2$ cuprate into the $J_{eff} = 1/2$ state of the iridate has suggested that electron-doping $Sr_2IrO_4$ is comparable to hole-doping $La_2CuO_4$[9–12], where an unconventional superconducting state is known to manifest[7,8].

Although bulk superconductivity has not yet been realized[13,14], a number of recent studies have uncovered an evolution of electronic phases in electron-doped $Sr_2IrO_4$ that mimic those observed in hole-doped $La_2CuO_4$. Surface doping studies have revealed the emergence of Fermi arcs and a $d$-wave quasiparticle gap with a well-defined onset temperature[15–17]. Similarly, studies of bulk electron-doped $Sr_2IrO_4$ have observed the opening of a pseudogap feature beyond a critical doping[18], and resonant inelastic scattering studies have established that robust magnon excitations persist far beyond the collapse of long-range antiferromagnetic order[19,20]. While these similarities suggest that electron-doping into the Mott state of $Sr_2IrO_4$ generates features reminiscent of hole-doped $La_2CuO_4$, the relative mapping of electron and hole concentrations between the two systems remains problematic. This is due to the fact that the stabilities of the two parent Mott states differ appreciably with respect to electron/hole doping.

While in hole-doped $(La_{1-x}Sr_x)_2CuO_4$ the insulating Mott state is completely suppressed by ~3% holes/Cu[7,8], in bulk electron-doped $(Sr_{1-x}La_x)_2IrO_4$ the Mott state remains only partially quenched at the highest doping levels currently achievable (~12% electrons/Ir)[13,14]. As a result, $(Sr_{1-x}La_x)_2IrO_4$ remains in a nanoscale electronically phase-separated ground state where insulating and metallic regions coexist[13,21]. While the insulating patches are remnants of the Mott state that support short-range antiferromagnetic correlations, a key open question concerns whether the metallic puddles also support a broken symmetry state such as the spin density wave states observed in the single-layer cuprates.

Here we address this possibility by presenting a high-resolution resonant elastic X-ray scattering (REXS) study exploring the evolution of magnetic order in electron-doped $(Sr_{1-x}La_x)_2IrO_4$. As the Mott state is suppressed and the material is driven into an electronically phase-separated regime, we observe the collapse of long-range antiferromagnetism and the appearance of an additional magnetic state that stabilizes coincident with the formation of a coherent Fermi surface in this system. Our data unveil a spin density wave phase with a character suggestive of the incommensurate spin density wave state known to stabilize in the metallic regions of far underdoped, electronically phase-separated $(La_{1-x}Sr_x)_2CuO_4$[8,22,23]. This commonality demonstrates universality in the electronic responses of the partially quenched Mott states of the monolayer hole-doped cuprates and electron-doped iridates.

## Results

**Magnetic order in $(Sr_{1-x}La_x)_2IrO_4$.** The evolution of magnetic order as electrons are introduced into $(Sr_{1-x}La_x)_2IrO_4$ is

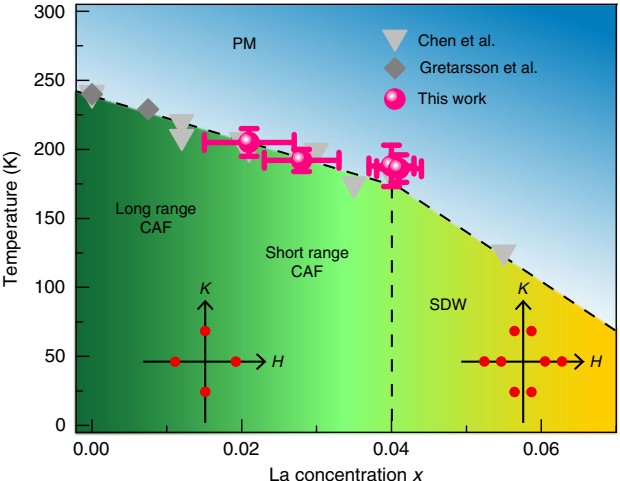

**Fig. 1** Phase diagram of $(Sr_{1-x}La_x)_2IrO_4$, as determined from a combination of magnetization and neutron scattering[13], RIXS[19], and REXS (this issue) measurements. The long-range to short-range canted antiferromagnetic (CAF) transition occurs near $x = 0.02$. At the critical doping $x = 0.04$, part of the material transitions into an incommensurate spin density wave (SDW) state. Dash lines are guides to eyes. The insets with red dots schematically represent commensurate and incommensurate (IC) peak positions investigated in this experiment. Horizontal error bars originate from repeated energy dispersive spectroscopy (EDS) measurements. Vertical error bars are estimated from the disappearance of magnetic peak intensity in Figs. 2d and 4e and Supplementary Figure 2a

summarized in Fig. 1. Previous reports have identified that, beyond a critical concentration of 2% electrons/Ir, long-range magnetic order collapses[19] and short-range, remnant order survives up to the solubility limit of La into the lattice[13,19]. While previous resonant inelastic X-ray (RIXS) measurements have observed the rapid formation of diffuse short-range magnetic correlations at electron-doping levels of 2% electrons/Ir[19], our REXS data collected near the Ir $L_3$ edge shown in Fig. 2 demonstrate that some fraction of the sample retains long-range antiferromagnetic (AF) order up to 6% electrons/Ir. This is not unexpected given the first order nature of the insulator to metal transition out of the Mott state and the known electron phase separation in this doping regime.

To further illustrate this remnant long-range order, Fig. 2(a–d) show diffraction data collected on two samples with La concentrations $x = 0.02$ and $x = 0.028$ at $T = 10$ K in the $\sigma - \pi$ scattering channel at the AF ordering wave vector. For reference, the $\mathbf{Q} = (0, 1, 4N + 2)$ and $(1, 0, 4N)$ type magnetic peaks of the parent state access one magnetic domain[3,24–26], while the $\mathbf{Q} = (0, 1, 4N)$ and $(1, 0, 4N + 2)$ positions access another. Consistent with earlier neutron diffraction studies[13], sharp, resolution-limited peaks in the $x = 0.02$ sample indicate that a small fraction of long-range order survives in the sample and that the ordering temperature decreases slightly with increased doping. As the doping is increased to $x = 0.028$, this remnant AF order broadens slightly along the $L$-axis with a finite out-of-plane correlation length $\xi = 320$ Å (Supplementary Figure 4) and presages the transformation to short-range magnetic order. As the doping level increases beyond this value, the long-range component of the remnant AF state is fully quenched at the commensurate $\mathbf{Q} = (0, 1, 14)$ wave vector; however, as additional electrons suppress this commensurate signal, they also drive the formation of a second set of peaks split along the $H$-axis from the AF wave vectors.

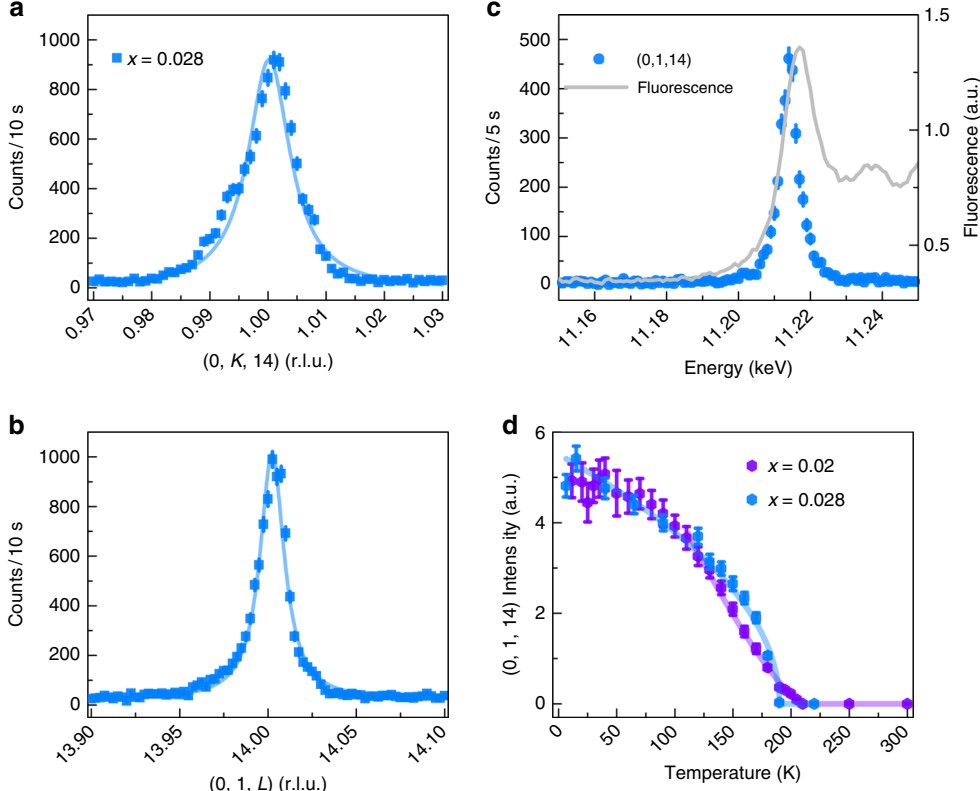

**Fig. 2** REXS data in the polarization flipped $\sigma$-$\pi$ channel of sample $x = 0.02$ (purple) and $x = 0.028$ (blue). **a** and **b** show $K$ and $L$ scans through the (0, 1, 14) position at $T = 10$ K respectively. **c** Energy dependence of the (0, 1, 14) peak at 10 K and its comparison to the Ir $L_3$ fluorescence line. **d** Temperature dependence of the (0, 1, 14) peaks for both samples. Data have been normalized to appear on the same scale. The solid lines in (**a**, **b**, **d**) are the fits to the data. Data are collected at the energies of 11.217 and 11.215 keV for the $x = 0.02$ and $x = 0.028$ sample, respectively. All vertical error bars in the figure represent 1 standard deviation (s.d.) statistical errors

Figure 3 illustrates the appearance of this additional channel of incommensurate scattering in a crystal with La concentration $x = 0.04$. Data in Fig. 3a and b depict scattering meshes collected about the charge position $\mathbf{Q} = (0, 0, 16)$ in the $\sigma$–$\sigma$ scattering channel and about the magnetic position $\mathbf{Q} = (0, 1, 14)$ in the $\sigma$–$\pi$ scattering channel. While the mesh about the charge peak reveals only a single-crystal grain, the mesh about the magnetic position reveals two incommensurate peaks split transversely along the $H$-direction in addition to the short-range commensurate AF peak. This is illustrated via momentum scans cutting through (0, 1, 4$N$ + 2) magnetic zone centers (Fig. 3c). These additional peaks are absent in the $\sigma$–$\sigma$ scattering channel (Fig. 3c) and appear below a temperature $T = 188 \pm 10$ K (Supplementary Figure 2a) coincident with the onset of the remnant short-range AF order in this system[13]. Momentum scans performed along the $L$-axis at the commensurate and incommensurate in-plane positions in Fig. 3d reveal that both peaks are quasi two-dimensional with finite out-of-plane correlation lengths of $\xi_c = 59 \pm 3$ Å (approximately 2 unit cells) for the remnant AF order of the parent state and a longer $\xi_c \approx 200$ Å (~8 unit cells) for the newly stabilized incommensurate scattering. The incommensurate scattering remains nearly resolution limited in the ($H$, $K$)-plane with a minimum correlation length of $\xi_{a,min} = 500 \pm 30$ Å.

Notably, the incommensurate peaks are asymmetrically split from the commensurate peak position along the $H$-axis at positions ($\delta$, 1, 14) with $\delta_1 \approx 0.001$ r.l.u. and $\delta_2 = -0.0035$ r.l.u. Mesh scans collected at charge peak positions such as that shown in Fig. 3a show that this anomalous splitting is not due to a

trivially misoriented grain in the sample and suggest the appearance of magnetic scattering with a different periodicity than the underlying lattice structure, coincident with the appearance of a Fermi surface at $x = 0.04$ as reported in previous photoemission experiments[18]. Maps exploring a magnetic zone center of the second allowed magnetic domain at the (1, 0, 10) position are shown in Fig. 3f. They reveal only one broadened, resolution-convolved peak where the small incommensurability is now largely masked by the asymmetric resolution function of the spectrometer. Though the resolution projected along $H$ in this geometry is 0.0023 r.l.u., analysis of a [$H$, $K$, 10] map (Supplementary Figure 3(b–d)) suggests this single peak is comprised of two components convolved together with an estimated splitting of $\delta_1 + \delta_2 = 0.005$ r.l.u. This demonstrates an inherent unidirectional splitting of the incommensurate scattering only along the $H$-direction, consistent with the orthorhombic symmetry of the underlying parent magnetic state.

Energy scans at both the commensurate and incommensurate positions are shown in Fig. 3e where both commensurate and incommensurate scattering components exhibit a well-defined resonance near the Ir $L_3$ edge. The resonance energy of the incommensurate signal is subtly shifted 1.5 eV lower than that of the short-range commensurate order and reflects resonant scattering from a different local electronic state within the sample. Such a negative shift in energy is consistent with the reported 1.3 eV shift downward in energy upon adding an electron into the $Ir^{4+}$ state and is likely reflective of a local $Ir^{3+}$ environment[27]. While this absolute energy difference is within the energy resolution of the monochromator, the subtle, relative

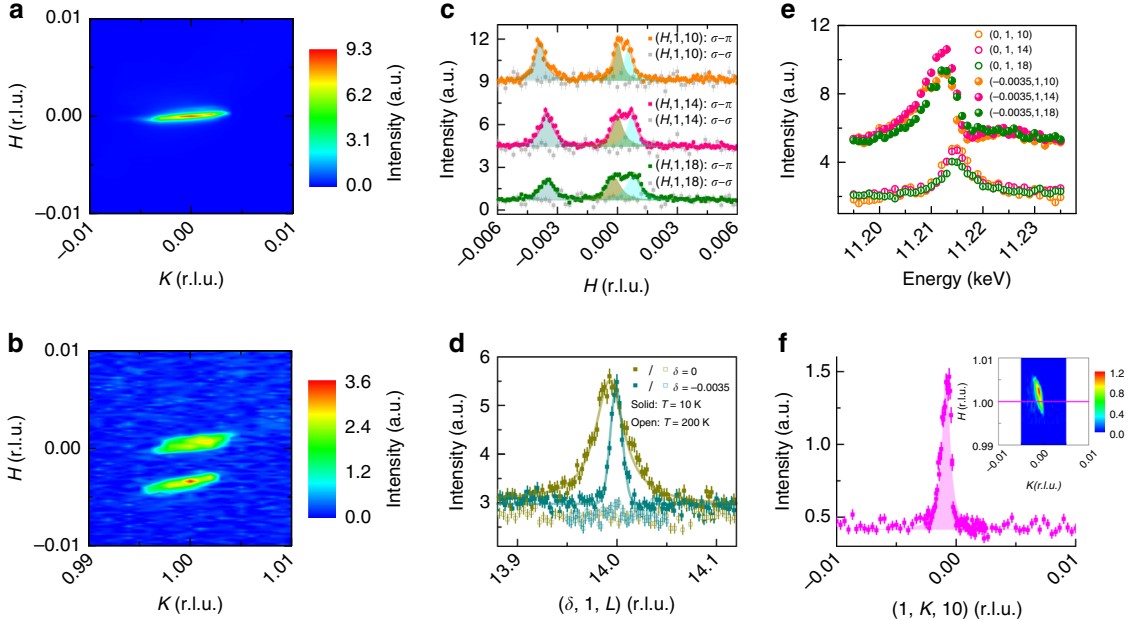

**Fig. 3** REXS data from sample $x = 0.04$ collected at 6-ID-B. **a**, **b** $H,K$ maps at the $(0, 0, 16)$ and $(0, 1, 14)$ zones at $T = 10$ K. **c** $H$ scans at select $(0, 1, 4N + 2)$ ($N = 2, 3, 4$) magnetic zone centers at 10 K collected both in the $\sigma$–$\pi$ channel and the $\sigma$–$\sigma$ channel. Solid lines are fits to the data with three Lorentzian peaks, which are individually represented by blue tan and cyan shaded areas. Data collected at different zones and in different channels are offset for comparison. **d** $L$ scans at the commensurate $(0, 1, 14)$ and incommensurate $(-0.0035, 1, 14)$ positions at 10 K (solid symbols) and at 200 K (empty symbols). The solid lines are fits to the data with a single Lorentzian peak and the dashed line denotes the instrumental resolution. **e** Energy scans at the commensurate and incommensurate peak positions. Data are offset for clarity. **f** $K$ scan at $(1, 0, 10)$ at $T = 10$ K after the sample was rotated counter-clockwise by 90°. The inset shows the $H,K$ map about the $(1, 0, 10)$ position and the magenta line denotes the scan direction in the main panel. No splitting along $K$ is observed, and additional analysis suggests a splitting still occurs along the $H$ direction (Supplementary Figure 3(b–f)). Other than (**e**), all data are collected at the energy of 11.214 keV. All vertical error bars in the figure represent 1 s.d. statistical errors

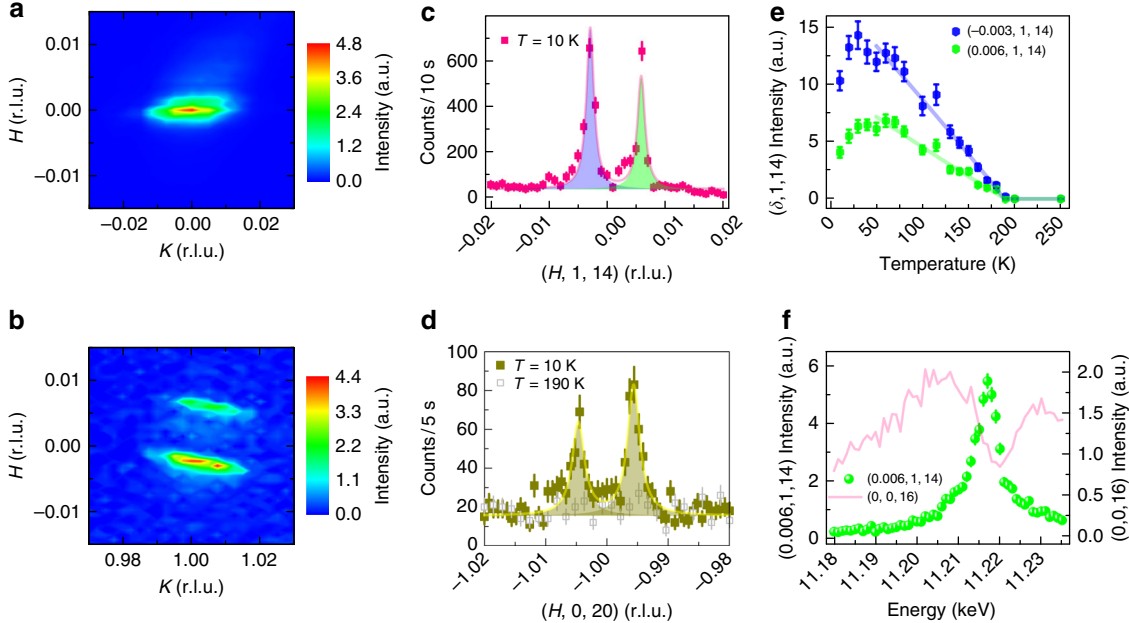

**Fig. 4** REXS data from sample $x = 0.041$ taken at CHESS, A2. **a** and **b** show $H,K$ maps $T = 10$ K around the nuclear $(0, 0, 16)$ and magnetic $(0, 1, 14)$ zones respectively. **c** $H$ scan at $(0, 1, 14)$ at 10 K exhibiting two IC peaks at $(-0.003, 1, 14)$ and $(0.006, 1, 14)$. **d** $H$ scan at $(-1, 0, 20)$ showing the two IC peaks split along the $H$ direction 10 K and their absence at 190 K. IC peaks in **c** and **d** are both separated by 0.009 r.l.u. **e** The temperature dependence of the two IC peaks. Both IC peaks disappear at the same temperature $T_{SDW} = 182 \pm 8$ K. **f** Energy dependence of the IC peak at $(0.006, 1, 14)$ and its comparison to the $(0, 0, 16)$ charge peak. The solid lines in **c**–**e** are the fits to the data. Other than **f**, all data are collected at the energy of 11.217 keV. All vertical error bars in the figure represent 1 s.d. statistical errors

shift is suggestive of resonant scattering from metallic regions of the sample where the Mott gap has collapsed.

To probe this new channel of order further, a separate sample with slightly higher La concentration $x = 0.041$ and better chemical homogeneity was also investigated with the results shown in Fig. 4. Scattering from this sample reveals only the presence of asymmetric incommensurate peaks split along $H$ about the $(0, 1, 4N + 2)$ positions and reflects a suppressed contribution from the remnant, competing short-range commensurate AF order. Figure 4(a,b) illustrate this via a mesh scan collected near the Ir $L_3$ edge in the $\sigma$–$\pi$ channel as well as via momentum scans illustrating the two peaks of magnetic scattering. With the removal of the remnant, competing short-range AF phase, the asymmetric incommensurate splitting is enhanced to $\delta_1 = -0.003$ r.l.u. and $\delta_2 = 0.006$ r.l.u. (Fig. 4c), and momentum scans exploring the same magnetic domain at the $(-1, 0, 20)$ position again demonstrate the inherent splitting only along the $H$-direction (Fig. 4d). Scans along the $L$-axis of the incommensurate peaks of this sample revealed an out-of-plane correlation length comparable to that observed in the mixed-phase sample (Supplementary Figure 4) and the in-plane correlation lengths remain resolution limited.

## Discussion

The emergent incommensurate electronic order observed in our scattering measurements for $x \geq 0.04$ coincides with the heterogeneous collapse of the Mott gap at this same doping. Prior scanning tunneling spectroscopy measurements[13,21] have shown that, while the local electronic structure is largely unperturbed for $x < 0.04$, at $x \approx 0.04$ dopants begin to locally stabilize a glassy pseudogap state and unidirectional electronic order within a nanoscale, phase-separated setting. The incommensurate state observed in our REXS measurements necessarily coexists with a background of diffuse magnetic scattering from the suppressed AF parent state previously reported in RIXS studies[19]. While previous RIXS measurements integrate larger swaths of momentum space and resolve this diffuse signal more readily, our complementary high-resolution REXS measurements are able to resolve the appearance of quasi two-dimensional order associated with the growing pseudogap phase fraction of the doped system. For higher La concentrations with $x \approx 0.05$, our measurements were unable to resolve any signatures of the spin density wave state. This is potentially due to a dramatic intensity reduction associated with the order becoming fully two-dimensional (diffuse along $L$) or due to a further suppression of the ordered magnetic moment as the metallic fraction of the sample grows.

At the limit of La-doping into $Sr_2IrO_4$, metallic nanopuddles comprise the bulk of the sample volume and are known to exhibit a pseudogap along antinodal regions considered analogous to quasiparticle spectra observed in the high-$T_c$ cuprates[18]. Similarly, hole doping into $Sr_2IrO_4$ also reveals parallel phenomenology such as the pseudogap and hidden order in cuprates[28-31]. The underlying nature of this pseudogap state is currently debated; however, the onset of short-range magnetic order is proposed as one possible origin. The quasi two-dimensional incommensurate order observed in our measurements is consistent with an electronic order parameter that may account for this pseudogap state; however, the detailed temperature dependence of the pseudogap phase remains unreported and currently limits further comparison.

Intriguingly, our measurements of $(Sr_{1-x}La_x)_2IrO_4$ at doping concentrations where a Fermi surface emerges seemingly parallel the appearance of a phase-separated spin density wave state in the analogous hole-doped cuprate $La_{2-x}Sr_xCuO_4$. Within the cuprate system, the parent Mott state along with long-range Néel order

rapidly collapse with hole substitution, and a competing unidirectional diagonal spin density wave (DSDW) state emerges[8,22,23]. This DSDW order stabilizes at the phase-separated limit of small doping and coexists with short-range commensurate AF order. The incommensurabilty inherent to the DSDW scales with the effective doping of the system with the smallest observed $2\delta \sim 0.016$ r.l.u.[32], and the propagation vector is split transverse to the commensurate AF wave vector—denoting a spin density wave modulation along the bond diagonal. Furthermore, at the low doping limit, coexistence of long-range DSDW correlations and AF order is observed in $La_{2-x}Sr_xCuO_4$ ($x = 0.0192$)[33] and ($x = 0.01$)[34]. These features parallel those observed for the competing incommensurate state in $(Sr_{1-x}La_x)_2IrO_4$ and are suggestive of a common instability in the phase diagrams of both systems. Given that the incommensurability in La-doped $Sr_2IrO_4$ is smaller than that observed in $La_{2-x}Sr_xCuO_4$ ($x = 0.01$), $(Sr_{1-x}La_x)_2IrO_4$ seemingly maps into the far under-doped regime of the same phase diagram where the in-plane correlation lengths remain long range.

Recent reports of the DSDW state at the low-doping limit of $La_{2-x}Sr_xCuO_4$ found a similar asymmetry in the incommensurate scattering[33]. The likely origin for the asymmetry in the incommensurability of doped $(Sr_{1-x}La_x)_2IrO_4$ is from a subtle orthorhombicity in the lattice below our current experimental resolution. A subtle shift of 0.0015 r.l.u. would rectify the scattering to be symmetric about the zone center, and this offset can readily be generated by an undetected orthorhombicity and twin domain structure within the lattice when aligned using a tetragonal cell. Since the spin modulation reflects a unidirectional density wave, the underlying lattice is necessarily orthorhombic or lower symmetry. A similar asymmetric incommensurate splitting was observed in hole-doped monolayer cuprates when the data is analyzed in a tetragonal cell[35], and for the small offset observed in our experiments, this would imply a rotation of twin domains by only 0.085° and an underlying orthorhombicity of $(a - b)/(a + b) = 0.001499$. Future high-resolution measurements of the domain structures in $(Sr_{1-x}La_x)_2IrO_4$ will be required to verify such a scenario in the monolayer iridates.

The mechanism for generating the incommensurate magnetic state for $x \geq 0.04$ at the edge of the Mott state's stability is likely to parallel previous proposals of a $t$-$t'$-$J$ model in the cuprates where, in the low doping limit, hopping of carriers can be maximized by a renormalized magnetic ground state[36-38]. Within this localized approach, Néel order in the Mott state becomes unstable above a critical doping threshold and twists into a spiral or noncollinear spin texture whose pitch evolves with carrier density. Details regarding the critical doping threshold and propagation wave vector of this modulated spin state can be material dependent, and notably, the introduction of appreciable Dzyaloshinskii-Moriya (DM) interactions in the case of electron-doped $Sr_2IrO_4$ will further modify the effect. While itinerant effects such as Fermi surface nesting cannot be completely ruled out, the spin modulation observed shows an inherently small incommensurability and no well-defined nesting wave vectors have been reported that would support such a state[18]. Further theoretical study of the stability of AF order in a $J_{eff} = 1/2$ Mott state with large DM interactions under light carrier doping will be required to more fully explain the origin of the DSDW state in $Sr_2IrO_4$.

While the microscopic origin of the DSDW state in $Sr_2IrO_4$ and the details of its domain structure remain open questions, our data directly reveal the presence of a competing magnetic order parameter which stabilizes in the metallic regime of $(Sr_{1-x}La_x)_2IrO_4$ and whose momentum space structure is reminiscent of the intermediate DSDW state of the high-$T_c$ cuprates. DSDW order is the magnetic precursor to superconductivity in the monolayer

cuprates and, given this analogy, our data suggest that electron-doped $(Sr_{1-x}La_x)_2IrO_4$ ($x \geq 0.04$) is on the verge of a bulk superconducting state. The emergence of a competing spin density wave upon electron-doping $(Sr_{1-x}La_x)_2IrO_4$ establishes a common feature in the collapse of the Mott states of the monolayer cuprates and iridates and motivates a push for higher electron concentrations as a route to realizing superconductivity in $(Sr_{1-x}La_x)_2IrO_4$.

## Methods

**Sample preparation.** Single crystals were grown via a platinum (Pt) crucible-based flux growth method as reported earlier[13]. Stoichiometric amounts of $SrCO_3$ (99.99%, Alfa Aesar), $La_2O_3$ (99.99%, Alfa Aesar), $IrO_2$ (99.99%, Alfa Aesar), and anhydrous $SrCl_2$ (99.5%, Alfa Aesar) were weighed in a $2(1-x)$: $x$: 1: 6 molar ratio, where $x$ is the nominal La concentration. The starting powders were fully ground, mixed and placed inside a Pt crucible, capped by a Pt lid, and further protected by an outer alumina crucible. Mixtures were heated slowly to 1380 °C, soaked for 5 to 10 h, slowly cooled to 850 °C over 120 h and then furnace cooled to room temperature over 5 h. Single crystals were then obtained after dissolving excess flux with deionized water.

**Sample characterization.** The samples studied in this paper were checked by a PANalytical Empyrean X-ray diffractometer at room temperature to exclude any possible $Sr_3Ir_2O_7$ phase. The La-doping concentrations were determined individually via energy dispersive spectroscopy (EDS) measurements with a typical spot size of $20 \times 20$ μm$^2$. The two $x \approx 0.04$ samples were further checked with different spot sizes, ranging from $0.5 \times 0.5$ to $500 \times 500$ μm$^2$. The concentrations observed are consistent within each sample regardless of the spot size chosen, indicating homogenous La content.

**Synchrotron X-ray scattering experiments.** REXS experiments were carried out at the A2 (samples with $x = 0.02$ and $x = 0.041$) and C1 ($x = 0.028$) beamlines at the Cornell High Energy Synchrotron Source, and the 6-ID-B beamline ($x = 0.04$) at the Advanced Photon Source at Argonne National Laboratory. Samples were mounted on the top of a Cu post and secured with GE varnish. A vertical scattering geometry was used with samples aligned in the (H0L) or (0KL) scattering planes. The experimental set-up is shown schematically in Supplementary Figure 1. The data were collected near the Ir $L_3$ edge ($E = 11.215$ keV, Supplementary Table I). Si (111) single crystals and NaI detectors were employed at the C1 and 6-ID-B beamlines. At the A2 beamline, scattered photon energy and polarization were analyzed using the symmetric (0, 0, 8) reflection from a flat HOPG analyzer crystal, and collected using a small area detector. The X-ray beam was vertically focused, and the incoming beam was horizontally polarized. Unless otherwise specified, all REXS data are collected in the polarization flipped $\sigma$–$\pi$ channel. The momentum transfer $\mathbf{Q} = (H, K, L)$ in scattering data is denoted in reciprocal lattice units, i.e., $\mathbf{Q} = (2\pi H/a, 2\pi K/b, 2\pi L/c)$, where $a = b \approx 5.5$ Å, $c \approx 25.8$ Å. All energy scans were performed at fixed $\mathbf{Q}$.

**Data availability.** All data are available from the corresponding author upon request.

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

## Acknowledgments

We thank Dr Gang Wu help for assistance with X-ray single-crystal measurements and structural refinement. This work was supported by NSF Award No. DMR-1505549 (X.C. and S.D.W.). Additional support was provided by ARO Award W911NF-16-1-0361 (J.S.,

E.Z., and Z.P.). The MRL Shared Experimental Facilities are supported by the MRSEC Program of the NSF under Award No. DMR 1121053; a member of the NSF-funded Materials Research Facilities Network. This research used resources of the Advanced Photon Source, a U.S. Department of Energy (DOE) Office of Science User Facility operated for the DOE Office of Science by Argonne National Laboratory under Contract No. DE-AC02-06CH11357. This work is based upon research conducted at the Cornell High Energy Synchrotron Source (CHESS) which is supported by the National Science Foundation and the National Institutes of Health/National Institute of General Medical Sciences under NSF award DMR-1332208.

## Author contribution

X.C. synthesized and characterized the $(Sr_{1-x}La_x)_2IrO_4$ single crystals. X.C. analyzed the data. X.C. perform the X-ray scattering experiment with the help from J.S., Z.I., Z.P., E.Z., K.F., J.P. C.R. X.C. and S. D. W. designed the experiments and prepared the manuscript with the input from all other co-authors.

## Additional information

**Competing interests:** The authors declare no competing financial interests.

