## [Peer Review File · Nature Communications]

Reviewers' comments:

Reviewer #1 (Remarks to the Author):

The authors studied the magnetism evolution of doped iridate $(\text{Sr}_{1-x}\text{La}_x)_2\text{IrO}_4$ in which Mott and metallic states coexist in nanoscale. They claim to observe a new incommensurate spin density wave in the metallic state, and by comparing with hole-doped $(\text{La}_{1-x}\text{Sr}_x)_2\text{CuO}_4$, they conjecture that both Mott quenched systems share common electronic response to carrier doping.

The studied topic is quite interesting, and their findings are potentially very important for understanding the high- T_c superconductivity in general. However, I think there are some serious questions that should be well addressed first.

(1) Single crystal refinement data for the experimental samples should be put in the supplementary material, in order to justify the crystalline quality. Also, is there observable lattice change with doping?

(2) It seems the magnetic peaks around $(0, 1, 14)$ are too sharp. For example, in Fig. 4, it looks the magnetic peak is even sharper than the $(0, 0, 16)$ lattice peak. This is rather anomalous, because the electronic order peaks should be broader than the lattice one. For instance, in $\text{La}_{2-x}\text{Sr}_x\text{CuO}_4$ (PRB, 65, 064505), the magnetic peaks are much broader (shorter correlation length/puddles) than its lattice peak. Also keep in mind that in this studied case there is also nanoscale electronic phase separation which should further broaden the electronic order peaks. The authors should overplot the $(0, 1, 14)$ magnetic peak with its $(0, 0, 16)$ lattice peak in a same figure in order to compare their peak width, and also compare with the magnetic peak in $\text{La}_{2-x}\text{Sr}_x\text{CuO}_4$, calculate the in-plane correlation length, and make a discussion.

(3) The asymmetric splitting in Fig. 3(b) and Fig. 4(b) are very strange. I wonder is it a real splitting, or just another incommensurate peak appearing aside the original magnetic peak. The authors should carefully exclude this possibility.

(4) How to explain the formation of incommensurate order from its electronic structure/Fermi surface? There should be theoretical support and/or related discussions on this observation.

(5) In Fig. 4 (e), there is a drop around 50 K, are there explanations for this drop?

Reviewer #2 (Remarks to the Author):

Chen et al. provide compelling evidence for an incommensurate CDW state emerging in electron doped Sr₂IrO₄. This is an important result, providing an additional and key piece of evidence that the physics of Sr₂IrO₄ exhibits some of the same underlying physics as the cuprate superconductors. Although the manuscript leaves many unanswered questions and thus motivates follow up work (additional dopings, further systematics, additional peaks, ...), I find this initial observation to be of sufficient significance to warrant publication in Nature Communications.

Prior to publication, I have a few comments/concerns that should be addressed.

The positions of the Bragg peaks, whose incommensurability shift with Q (F3c) and their asymmetry in + or – H are somewhat alarming and cast some doubt on the veracity of the measurements. One possibility to explain some of these effects may be refraction. With resonant x-ray scattering the index of refraction is photon energy (and photon polarization) dependent. This can lead to shifts in the apparent Q of Bragg peaks as a function of photon energy as the angle of incident and scattered beams deviate at the surface of the crystal. See the supplementary information of Achkar et al. Science 351, 576 (2016) for an example. The authors should consider where this effect can explain the measured peak positions and asymmetry, including Fig. S2e,f.

Related to this concern, the photon energy that each all scans are taken at should be clearly indicated in the figures or captions. Moreover, the authors should specify how the energy scans (Figs 2c and 3e) were performed. Are they scans of photon energy at a fixed angle?

Minor corrections:

Typo: Supp. Fig. 2 Caption, energies “E = 209 keV (dark cyan) and E = 217 keV” should be 11.209 keV and 11.217 keV

Some references are incomplete, lacking article numbers. These should be checked.

Reviewer #3 (Remarks to the Author):

The manuscript by Chen et al. report a study of the evolution of the magnetic structure in Sr₂IrO₄ with La (electron) doping.

Sr₂IrO₄ is of considerable current interest as it exemplifies the new class of spin-orbit Mott insulators.

One particularly important open question is to what extent the electronic/magnetic state in Sr₂IrO₄ differs from that in La₂CuO₄, with relevance for high-T superconductivity, etc.

Chen et al. report the results of resonant magnetic X-ray diffraction experiments performed to understand

how La (electron) doping modifies the commensurate antiferromagnetic order of the parent Sr₂IrO₄ compound.

The main claim made in the paper is the observation around $x=0.04$ of a previously unreported incommensurate phase. They argue that this phase is a spin-density wave state similar to the diagonal stripe phase observed in lightly hole doped La₂IrO₄ that presages the onset of superconductivity at higher doping.

This result therefore may be seen as uniting a key aspect of the phenomenology of two disparate families

of transition metal compounds - cuprates and iridates - which will be of interest to the large community of researchers working on these materials, and could be of potential wider appeal.

The experiments appear to have been performed to a high standard, and the manuscript itself is generally well written.

However, there is one aspect of the data and its interpretation which I see as being deeply problematical, and makes me hesitate from recommending the manuscript in its present form.

This relates to the fact that the additional peaks observed in the $x=0.04$ and 0.041 samples - ie the ones

on which the claim for the observation of a spin-density wave are predicated - do not occur at positions

that can be readily related to the crystallographic lattice (Fig. 3a,b,c and Fig. 4a-d).

The authors themselves notice this "anomaly" without being able to present a definitive explanation of

its origin. I find this to be a serious weakness of the current manuscript as it leaves open the possibility

that something is awry with the experiments. Could it, for example, be an artefact produced by

the
experimental setup? One such possibility might be that the analyser used for the polarization
analysis
has several crystallographic domains, but there are others that should be excluded.

An additional clue that something might be off with the experiments - in addition to a lack of
centering of
the putative incommensurate peaks - is the fact that the peaks are not really centred in the K
direction,
as can be seen from Figs. 3b and 4b.

I therefore do not recommend acceptance of the manuscript in its current form.
For me to do so will require that a more convincing explanation for the origin of the positional
"anomaly" is provided by the authors, including if necessary further diagnostic tests.

Reviewer #1 (Remarks to the Author):

The authors studied the magnetism evolution of doped iridate $(\text{Sr}_{1-x}\text{La}_x)_2\text{IrO}_4$ in which Mott and metallic states coexist in nanoscale. They claim to observe a new incommensurate spin density wave in the metallic state, and by comparing with hole-doped $(\text{La}_{1-x}\text{Sr}_x)_2\text{CuO}_4$, they conjecture that both Mott quenched systems share common electronic response to carrier doping.

The studied topic is quite interesting, and their findings are potentially very important for understanding the high- T_c superconductivity in general. However, I think there are some serious questions that should be well addressed first.

We thank the referee for his/her careful reading of the manuscript and for noting the general interest and importance of our results.

(1) Single crystal refinement data for the experimental samples should be put in the supplementary material, in order to justify the crystalline quality. Also, is there observable lattice change with doping?

We have now performed single crystal refinement for two of the key samples in our study as well as a parent ($x=0$) sample. The most important comparison is nominally between the $x=0$, $x=0.028$, and $x=0.04$ samples; however we note here that, unfortunately, we could not perform a refinement on the $x=0.041$ sample as it was lost during shipment back to UCSB from the CHESS synchrotron. The results from these refinements have been added to the supplemental materials in order to justify the crystal quality of samples studied. We note that there is no apparent change in crystal quality from the $x=0.028$ to $x=0.04$ samples where the transition to the incommensurate spin state occurs. Also for the referee's background reference, lattice changes in La-doped Sr_2IrO_4 are small due to counteracting steric effects (La^{3+} is smaller than Sr^{2+}) and electron deformation potential effects (adding electrons in the conduction band lower their energy by expanding the lattice). The current data is consistent with our previous higher resolution reports on crushed single crystals and powders in Phys. Rev. B **92**, 075125 (2015).

(2) It seems the magnetic peaks around (0, 1, 14) are too sharp. For example, in Fig. 4, it looks the magnetic peak is even sharper than the (0, 0, 16) lattice peak. This is rather anomalous, because the electronic order peaks should be broader than the lattice one. For instance, in La_{2-x}Sr_xCuO₄ (PRB, 65, 064505), the magnetic peaks are much broader (shorter correlation length/puddles) than its lattice peak. Also keep in mind that in this studied case there is also nanoscale electronic phase separation which should further broaden the electronic order peaks. The authors should overplot the (0, 1, 14) magnetic peak with its (0, 0, 16) lattice peak in a same figure in order to compare their peak width, and also compare with the magnetic peak in La_{2-x}Sr_xCuO₄, calculate the in-plane correlation length, and make a discussion.

Fig 1: Comparison of magnetic and charge peak widths for the $x=0.04$ (a) and $x=0.041$ (b) samples. Upper right insets in each panel show the charge peak widths and these widths are over-plotted as dashed lines inside of the magnetic peaks for reference.

We have now provided a figure in the supplemental information directly comparing the peak widths of the magnetic (0, 1, 14) and the lattice peak (we reproduce this plot below for the reviewer's convenience). The nuclear peak in Fig 4a appears to be slightly broader because of the smoothing and interpolation of the 2d data. The line cuts show the incommensurate magnetic peaks (FWHM=0.0017(1) r.l.u.) are slightly broader than the nuclear peaks (FWHM=0.0016(2) r.l.u.), as is shown in the dashed lines of the figure (Fig I (b)). This slight broadening though is fairly close to the resolution of the measurement, and at this limit comparing the crystallinity and resolution-convolved widths of the charge peaks at different Q_s with the magnetic peak may give somewhat misleading correlation lengths. It is probably safest to simply provide minimum correlation lengths for the in-plane moments, which we have now included in the paper.

The reviewer also raises a question regarding the comparison of in-plane correlation lengths to the diagonal density wave state in LSCO. Notably, recent reports studying the far underdoped regime of La_{2-x}Sr_xCuO₄ ($x=0.0192$) (Gil Drachuck et al., Nature Communications 5, 3390 (2014) (DOI: 10.1038/ncomms4390)) see long-range diagonal spin density wave order within the plane and also on the same scale of the coexisting commensurate AF order. Interestingly, this study also reports a notable asymmetry in the long-wavelength modulated satellites of the DSDW state in far underdoped LSCO. This is in agreement with earlier studies by Matsuda et al. in Phys. Rev. B 65, 134515 (2002) where in LSCO ($x=0.01$)

the in-plane correlation length corresponding to our sharp (H) direction is also resolution limited (ie. Long-range). At slightly higher doping levels in LSCO ($x=0.024$), reports by Matsuda et al. Phys. Rev. B 62, 9148 (2000) see this broaden to be about 95 Å within the plane and this continues to broaden with increased doping as the referee noted. Given that the incommensurability that we observe in La-doped Sr₂IrO₄ is smaller than LSCO ($x=0.01$), we consider (Sr_{1-x}La_x)₂IrO₄ to be in the far under doped regime of the same phase diagram where the in-plane correlation lengths remain long range. One further thing to note is that, due to the small incommensurability, the correlation lengths are necessarily long in order for us to resolve the scattering before it becomes too diffuse to observe with the REXS technique.

We have now added this discussion to the manuscript text.

(3) The asymmetric splitting in Fig. 3(b) and Fig. 4(b) are very strange. I wonder is it a real splitting, or just another incommensurate peak appearing aside the original magnetic peak. The authors should carefully exclude this possibility.

From the systematics of our measurements we believe can exclude an additional incommensurate peak appearing alongside the original commensurate peak coming from the same domain. The different resonance energies of each peak suggest a different origin for the commensurate and incommensurate peaks in the $x=0.04$ sample, and in the $x=0.041$ sample (where the short-range commensurate order is no longer resolvable) both incommensurate peaks resonate at the same energy. As we will discuss at the end of our reply to reviewer #3, the asymmetric splitting is likely an apparent effect from the presence of two different twin domains.

(4) How to explain the formation of incommensurate order from its electronic structure/Fermi surface? There should be theoretical support and/or related discussions on this observation.

We have now added a discussion to the text regarding the potential origin of this incommensurate state. A likely model for understanding this magnetic state at the edge of the Mott state's stability is to think of things in terms of a t-J model where hopping can be maximized by a renormalized magnetic ground state. Such a model has been developed in the case of the hole-doped cuprates (PRL **98**, 037001 (2007), PRL **62**, 1564 (1989), PRB **41**, 2653 (1990)) and the essential picture is that, above a critical doping threshold, Neel order in the Mott state becomes unstable and forms a spiral state that allows enhanced hopping. The period of this spiral state evolves with carrier density, and we believe that similar physics may be operative here. However, we note that the details for the case of the iridates where DM interactions cannot be ignored may be slightly different. We have also added discussion of this predicted spiral state to the text. We note that, while we cannot completely rule out Fermi surface nesting, no well-defined nesting wave vectors have been found to support the incommensurate order (PRL 115, 176402 (2015)).

(5) In Fig. 4 (e), there is a drop around 50 K, are there explanations for this drop?

We believe this due to uncertainties in collecting and normalizing the order parameter. This does not repeat for instance in the $x=0.04$ sample, so we hesitate to ascribe any meaning to the apparent drop.

Reviewer #2 (Remarks to the Author):

Chen et al. provide compelling evidence for an incommensurate CDW state emerging in electron doped Sr2IrO4. This is an important result, providing an additional and key piece of evidence that the physics of Sr2IrO4 exhibits some of the same underlying physics as the cuprate superconductors. Although the manuscript leaves many unanswered questions and thus motivates follow up work (additional dopings, further systematics, additional peaks, ...), I find this initial observation to be of sufficient significance to warrant publication in Nature Communications.

We thank the referee for his/her review and support of the publication of our findings in Nature Communications.

Prior to publication, I have a few comments/concerns that should be addressed.

The positions of the Bragg peaks, whose incommensurability shift with Q (F3c) and their asymmetry in + or - H are somewhat alarming and cast some doubt on the veracity of the measurements. One possibility to explain some of these effects may be refraction. With resonant x-ray scattering the index of refraction is photon energy (and photon polarization) dependent. This can lead to shifts in the apparent Q of Bragg peaks as a function of photon energy as the angle of incident and scattered beams deviate at the surface of the crystal. See the supplementary information of Achkar et al. Science 351, 576 (2016) for an example. The authors should consider where this effect can explain the measured peak positions and asymmetry, including Fig. S2e,f.

The referee raises a good question here regarding the potential for dynamical diffraction effects in accounting for an offset in the scattering position of the peaks. These effects are typically larger in soft-ray measurements (ie. 3d transition metal absorption edges), nevertheless, we endeavored to try and simulate this effect in our experiments. We simulated these effects using the QUAD simulation package (Phys. Rev. Lett. 117, 115501) where the charge scattering amplitudes for Sr and O are obtained from tabulated data ($f_{Sr} = 36 + 0.9i$ and $f_O = 8 + 0.0095i$ at energy $E = 11.215$ keV) by B.L. Henke et al. At. Data Nucl. Data Tables 54, 181–342 (1993) and the charge and magnetic scattering amplitudes for Ir are obtained from the optical theorem relating the imaginary part of the scattering amplitudes to the absorption data (XAS and XMCD data from D Haskel et al. PRL 109, 027204 (2012))

Fig II: Simulated peak positions for scattering at the (0,0,16) charge peak and (0,0,13) magnetic peak under the kinematic and dynamic approximations. Intensity is scaled for comparison.

and then through Kramers-Kronig transform which relates the real and imaginary parts of the scattering amplitudes ($f_{\text{Charge, Ir}} = -33 + 35i$, $f_{\text{Mag, Ir}} = 0.025 + 0.34i$ at energy $E = 11.215$ keV).

Our simulations do indicate that the dynamic effects shift the position of maximum intensity of Bragg peaks to offset positions; however such effects are quite small (only 0.00075 r.l.u. along L direction and even smaller along in plane directions, as in Fig II). These offsets are also largely Q independent—the charge peaks shift as much as the magnetic peaks. We feel that these small simulated shifts cannot fully explain the offset of incommensurate magnetic peaks which are estimated to be 0.0015 r.l.u. along H direction. Additionally, since the alignment is based on the charge Bragg peaks, if the dynamic effects affect both the charge and magnetic peaks the same way, the apparent dynamic shift would be even smaller.

Related to this concern, the photon energy that each all scans are taken at should be clearly indicated in the figures or captions. Moreover, the authors should specify how the energy scans (Figs 2c and 3e) were performed. Are they scans of photon energy at a fixed angle?

We have now added this information to the figures and the text. The energy scans were performed at a fixed Q, so the angle varies slightly with changing energy.

Minor corrections:

Typo: Supp. Fig. 2 Caption, energies “E = 209 keV (dark cyan) and E = 217 keV” should be 11.209 keV and 11.217 keV

Thank you for noting this. This has now been corrected.

Some references are incomplete, lacking article numbers. These should be checked.

This has now been fixed. Thanks for calling this to our attention.

Reviewer #3 (Remarks to the Author):

The manuscript by Chen et al. report a study of the evolution of the magnetic structure in Sr2IrO4 with La (electron) doping. Sr2IrO4 is of considerable current interest as it exemplifies the new class of spin-orbit Mott insulators. One particular important open question is to what extent the electronic/magnetic state in Sr2IrO4 differs from that in La2CuO4, with relevance for high-T superconductivity, etc.

Chen et al. report the results of resonant magnetic X-ray diffraction experiments performed to understand how La (electron) doping modifies the commensurate antiferromagnetic order of the parent Sr214 compound. The main claim made in the paper is the observation around $x=0.04$ of a previously unreported incommensurate phase. They argue that this phase is a spin-density wave state similar to the diagonal stripe phase observed in lightly hole doped La214 that presages the onset of superconductivity at higher doping.

This result therefore may be seen as uniting a key aspect of the phenomenology of two disparate families of transition metal compounds - cuprates and iridates - which will be of interest to the large community of researchers working on these materials, and could be of potential wider appeal.

The experiments appear to have been performed to a high standard, and the manuscript itself is generally well written.

We thank the referee for his/her careful reading of our manuscript and for noting the broad interest of the manuscript's conclusions.

However, there is one aspect of the data and its interpretation which I see as being deeply problematical, and makes me hesitate from recommending the manuscript in its present form.

This relates to the fact that the additional peaks observed in the $x=0.04$ and 0.041 samples - ie the ones on which the claim for the observation of a spin-density wave are predicated - do not occur at positions that can be readily related to the crystallographic lattice (Fig. 3a,b,c and Fig. 4a-d). The authors themselves notice this "anomaly" without being able to present a definitive explanation of its origin. I find this to be a serious weakness of the current manuscript as it leaves open the possibility that something is awry with the experiments. Could it, for example, be an artefact produced by the experimental setup? One such possibility might be that the analyser used for the polarization analysis has several crystallographic domains, but there are others that should be excluded.

We have now added an in-depth discussion of the possible origins of the asymmetric splitting in our sample along with text in the supplementary information which excludes extrinsic effects like mosaicity in the analyzer. For the referee's convenience, we produce a plot below of the analyzer scan at the (0, 2, 16) charge peak position for our CHESS measurements. Furthermore, many extrinsic spectrometer dependent effects can be excluded by noting that we see this incommensurate splitting on two different beamlines with different experimental configurations at 6-ID-B and A2. In addition, the low doping concentrations (such as the $x=0.02$ sample) collected on the same instruments using the same configurations did not show similar features.

Fig III: Analyzer Θ scan at the (0, 2, 16) charge peak.

An additional clue that something might be off with the experiments - in addition to a lack of centering of the putative incommensurate peaks - is the fact that the peaks are not really centred in the K direction, as can be seen from Figs. 3b and 4b.

The referee is correct in noting that there appears to be a small offset along the K-direction, although it is harder to quantify due to the poorer resolution in that direction. This offset in this direction is likely best understood via a combination of alignment uncertainties and an underlying orthorhombicity to the lattice (as we explain in the next point). For the referee's reference, we plot a map of a charge peak at the (0, 2, 16) for the $x = 0.041$ sample showing what we mean by alignment uncertainty in the poor resolution (K) direction (Fig IV). This charge peak map also appears slightly offset along K, although this is purely due to the extrinsic fact that the alignment and lattice are set via line scans along this broader direction and carries with it some uncertainty.

Fig IV: 2D in-plane intensity map of (0,2,16) at T = 10K for x = 0.041 sample.

I therefore do not recommend acceptance of the manuscript in its current form. For me to do so will require that a more convincing explanation for the origin of the positional "anomaly" is provided by the authors, including if necessary further diagnostic tests.

Following further analysis of our data, the most consistent explanation of the asymmetry we observe in the incommensurate peaks is that the magnetism reflects a subtle underlying orthorhombic symmetry of the lattice. Since the spin modulation we observe reflects a unidirectional density wave, by symmetry *the underlying lattice is necessarily orthorhombic or lower symmetry*. If this orthorhombicity is small, it can be very easy to miss due to lattice twinning effects (see for instance *Physical Review B* 93, 134110 (2016) for the difficulty in finding the inherent crystal structure in the related compound $\text{Sr}_3\text{Ir}_2\text{O}_7$). A similar conundrum (asymmetric incommensurate splitting) was encountered in hole-doped La_2CuO_4 when the data was first analyzed in a tetragonal cell, and there the breaking of the in-plane four-fold rotational symmetry provided the most physical interpretation.

A number of twinning scenarios are possible with typical in-plane orthorhombic twinning patterns of $(a_{\text{short}}, b_{\text{long}})$, $(-a_{\text{short}}, b_{\text{long}})$, $(b_{\text{long}}, a_{\text{short}})$, $(-b_{\text{long}}, a_{\text{short}})$. For our case, we illustrate how this can generate an apparent offset by considering a simple two-domain structure possible in an orthorhombic lattice. The figure below illustrates two domains possible in a subtly orthorhombic lattice:

Fig V: (Left panel) Tetragonal and orthorhombic axes for the basal IrO_2 planes. Tetragonal here denotes the smaller unit cell where H and K point along the Ir-O-Ir bond direction while orthorhombic denotes the larger unit cell with H and K parallel to the bond diagonals. Two potential domains A and B are marked if the lattice is subtly orthorhombic. (Right panel) H and K axes again in the larger cell with circles denoting the aligned (1,0) and (0,1) positions assuming $a_{\text{orth}}=b_{\text{orth}}$ and the relative orientations of the A and B domains under this assumption. Stars denote the unidirectional splitting resolution from each domain with modulation along the H_{orth} -axis.

Alignment of the charge peaks assuming a tetragonal lattice will roughly choose an average position between the two domains and generate an apparent offset for each domain's magnetic zone center using a tetragonal UB matrix. For the small offset we see in our experiments (0.0015 r.l.u.), this would imply a rotation of domains by only 0.085° and an underlying orthorhombicity of $(a-b)/(a+b)=0.001499$. Additionally, in this picture initial alignment uncertainties based on where we center in the tetragonal lattice would cause variance in the apparent peak positions between measurements (like in Figs. 3b and 4b), yet the offset required to fix the incommensurability should not change, which is what we see. So in this simple example, one could imagine resolving two peaks from domain A which would appear offset from the commensurate position in the $x=0.041$ sample.

In this simple picture, our measurements would resolve only either domain A or domain B due to a combination of the magnetic structure factor and the relative orientation of spins in each domain. For instance, only the (1,0) or (0,1) zone centers are allowed for $L=4N$ or $4N+2$ respectively if the the c-axis stacking of the canted spin structure of the parent AF order is preserved. This selection rule seems to remain intact for the locations of incommensurate peaks in the $x=0.04$ and $x=0.041$ samples. Discerning further selection rules would require more detailed knowledge on the lattice domain structure and the spin modulation and orientation within a given domain (e.g. sinusoidal versus cycloidal). Resolving these details will require a program of sustained high resolution scattering measurements, which we plan to perform; however we feel that the current report of the discovery of this incommensurate state stands on its own.

Reviewers' Comments:

Reviewer #1 (Remarks to the Author):

I found that the authors answer properly to most of my concerns. However, the interpretation about the asymmetry anomaly, as the most important part of this paper, is still based on speculation. A high resolution experiment for justification is necessary, but I understand that it takes a lot of experimental planning and beamtime applications etc. Since the current manuscript reports an important experimental finding of a incommensurate peak, no matter what it really is, I recommend the acceptance of the paper on the condition that the authors need to point out the unresolved experimental issues.

Reviewer #2 (Remarks to the Author):

The authors have adequately responded to the concerns of my initial report. Although there remain outstanding questions that need to be resolved regarding this newfound incommensurate density wave order in doped irridates, I find these initial observations sufficiently compelling to warrant publication in Nature Communications.

Reviewer #3 (Remarks to the Author):

The authors have addressed my concerns in a reasonable way, and I am happy to now recommend the manuscript for publication in NC.

Responses to Reviewers:

Reviewer #1 (Remarks to the Author):

I found that the authors answer properly to most of my concerns. However, the interpretation about the asymmetry anomaly, as the most important part of this paper, is still based on speculation. A high resolution experiment for justification is necessary, but I understand that it takes a lot of experimental planning and beamtime applications etc. Since the current manuscript reports an important experimental finding of a incommensurate peak, no matter what it really is, I recommend the acceptance of the paper on the condition that the authors need to point out the unresolved experimental issues.

We thank the referee for his/her continued review of our manuscript and for recommending its acceptance in Nature Communications. To address the referee's parting concern regarding the remaining open question about the origin of the asymmetry anomaly, we have now added a sentence to the discussion emphasizing that future experiments will be necessary to verify the origin of the anomaly.

Reviewer #2 (Remarks to the Author):

The authors have adequately responded to the concerns of my initial report. Although there remain outstanding questions that need to be resolved regarding this newfound incommensurate density wave order in doped irridates, I find these initial observations sufficiently compelling to warrant publication in Nature Communications.

We thank the referee for his/her careful review of our manuscript and for recommending its publication in Nature Communications.

Reviewer #3 (Remarks to the Author):

The authors have addressed my concerns in a reasonable way, and I am happy to now recommend the manuscript for publication in NC.

We thank the referee for recommending the manuscript's publication in Nature Communications.